# Neurotensin and Alcohol Use Disorders: Towards a Pharmacological Treatment

**DOI:** 10.3390/ijms24108656

**Published:** 2023-05-12

**Authors:** Francisco D. Rodríguez, Manuel Lisardo Sánchez, Rafael Coveñas

**Affiliations:** 1Department of Biochemistry and Molecular Biology, Faculty of Chemical Sciences, University of Salamanca, 37008 Salamanca, Spain; 2Group GIR-USAL: BMD (Bases Moleculares del Desarrollo), University of Salamanca, 37008 Salamanca, Spain; 3Laboratory of Neuroanatomy of the Peptidergic Systems, Institute of Neurosciences of Castilla and León (INCYL), University of Salamanca, C/Pintor Fernando Gallego 1, 37007 Salamanca, Spain

**Keywords:** alcohol use disorder (AUD), neurotensin, neurotensin receptors, neurotensin signaling, brain reward

## Abstract

Harmful alcohol use is responsible for a group of disorders collectively named alcohol use disorders (AUDs), according to the DSM-5 classification. The damage induced by alcohol depends on the amount, time, and consumption patterns (continuous and heavy episodic drinking). It affects individual global well-being and social and familial environments with variable impact. Alcohol addiction manifests with different degrees of organ and mental health detriment for the individual, exhibiting two main traits: compulsive drinking and negative emotional states occurring at withdrawal, frequently causing relapse episodes. Numerous individual and living conditions, including the concomitant use of other psychoactive substances, lie in the complexity of AUD. Ethanol and its metabolites directly impact the tissues and may cause local damage or alter the homeostasis of brain neurotransmission, immunity scaffolding, or cell repair biochemical pathways. Brain modulator and neurotransmitter-assembled neurocircuitries govern reward, reinforcement, social interaction, and consumption of alcohol behaviors in an intertwined manner. Experimental evidence supports the participation of neurotensin (NT) in preclinical models of alcohol addiction. For example, NT neurons in the central nucleus of the amygdala projecting to the parabrachial nucleus strengthen alcohol consumption and preference. In addition, the levels of NT in the frontal cortex were found to be lower in rats bred to prefer alcohol to water in a free alcohol–water choice compared to wild-type animals. NT receptors 1 and 2 seem to be involved in alcohol consumption and alcohol effects in several models of knockout mice. This review aims to present an updated picture of the role of NT systems in alcohol addiction and the possible use of nonpeptide ligands modulating the activity of the NT system, applied to experimental animal models of harmful drinking behavior mimicking alcohol addiction leading to health ruin in humans.

## 1. Introduction

Alcohol use disorders (AUDs) are a group of heterogeneous clinical entities and symptoms caused by harmful alcohol consumption. The damage induced by alcohol ingestion depends on the amount, time, and consumption patterns (continuous and heavy episodic drinking). It affects individual global well-being and social and familial environments with variable impact. Alcohol addiction manifests with different degrees of organ and mental health detriment for the individual, exhibiting two traits worth highlighting: compulsive drinking and negative emotional states occurring at withdrawal, frequently resulting in relapse episodes [1,2,3,4].

The World Health Organization (WHO), in the 2018 report on alcohol and health [5], briefs that approximately 2300 million individuals are current alcohol consumers. In addition, globally, harmful alcohol drinking was responsible for approximately three million deaths and 133 million disability-adjusted life years (DALYs). According to the WHO 2019 Status Report on alcohol consumption, harm, and policy responses in 30 European countries [6], alcohol caused 5.5% of all deaths in 2016 in this region. Alcohol-attributable deaths were due to cancer (29%), liver cirrhosis (20%), cardiovascular disease (19%), and injury (18%). In addition, the same study advises that, in 2016, alcohol use caused 8.3% of the YLL (years of life lost), representing 7.6 million years lost. In Spain, alcohol is the most consumed drug, representing the fourth risk factor of health deterioration (measured as DALYs), and directly caused approximately 15,000 deaths during 2007–2017. Approximately 56% of deaths were premature (before age 75) [7]. Most of the health burden related to excessive alcohol drinking, specifically deaths, affects individuals suffering from severe AUD [8]. No doubt, the consequences of harmful alcohol consumption on human health are remarkable. All parties (individuals, families, and society) need dedicated and persistent attention to achieve effective prevention and treatment procedures. The problems originating from harmful alcohol consumption must be tackled from a multidisciplinary perspective, including the legal aspects of drinking, social, education, prevention programs, and medical intervention. Pari passu, basic, and clinical research should proceed to advance new and effective solutions.

AUDs include various clinical entities influenced by alcohol consumption and individual and social–family factors. The principal etiological factor causing AUD is ethyl alcohol and its metabolites, which directly damage tissues and organs. Alcohol is a positive stimulus to the brain; however, it eventually disturbs its neurotransmission, generating maladaptation changes that perpetuate consumption and damage. The brain model of addiction (Figure 1) proposes a circle of consumption and responses where alcohol ingestion escalates from intermittent drinking producing pleasurable effects to consuming to alleviate the discomfort symptoms derived from abstinence [9,10,11,12,13]. Dopamine in the nucleus accumbens is a key neurotransmitter responsible for the immediate reinforcing outcomes induced by acute ethanol (and other drugs of abuse). In contrast, chronic alcohol exposure alters dopaminergic striate–thalamus–cortical and limbic connections, possibly leading to addiction in susceptible drinkers [14].

Brain peptidergic systems from distant outputs and local peptidergic neurons in specific brain regions (for example, ventral tegmental area, nucleus accumbens, the paraventricular nucleus of the thalamus, hypothalamus, extended amygdala, or prefrontal cortex) control dopamine, opioid, GABA (γ-amino butyric acid), and glutamate neurotransmission. Ethyl alcohol affects these neurotransmitter systems, and an addiction cycle characterized by neurotransmission’s maladaptation significantly perpetuates a vicious addiction circle [16,17,18,19,20,21].

The neuroscientific community generally views alcohol addiction (substance addiction) as a brain disease and not a chosen behavior option. In fact, in 2011, the American Society of Addiction Medicine defined addiction as a chronic brain disorder [22]. Consequently, this conception promotes studying etiological factors and molecular mechanisms as potential targets for pharmacological treatment [3]. However, this endeavor does not exclude soundly tackling the psychological, social, economic, and legal aspects and damaging consequences of addiction.

Despite existing officially approved drugs (although no silver bullet cure is available) to treat AUD, treatment rates are low, at approximately 16% [23]. The complexity of AUDs and the individual adaptations to the stimulus, alcohol, the social stigma attached to alcohol addiction, the low capacity of some individuals to access or stick to treatments, and in some cases, the lack of attention by qualified professionals may explain the low figure. In addition, AUD appears to be frequently associated with other abused substances, leading to co-morbid entities with worse prognoses [24]. Because of the variety of AUD entities, some individuals respond to specific treatments, others respond mildly, and others do not. Innovative programs to establish proper diagnosis and methods to engage patients should combine behavioral and pharmacological therapies, and consideration of the effect of individual genetic polymorphisms may lead to successful treatments [25]. Current approved pharmacological treatments include benzodiazepines for withdrawal treatment, disulfiram (an acetaldehyde dehydrogenase substrate suicide inhibitor), opioid antagonists (e.g., naltrexone), and acamprosate (a glutamate modulator) to maintain abstinence and nalmefene (an antagonist at the μ and δ-opioid receptors, and a partial agonist at the κ-opioid receptor) to reduce heavy alcohol consumption [24,25,26,27]. Other compounds with an encouraging application are topiramate (modulates GABA and glutamate activities), baclofen and gabapentin (GABA analogs), varenicline (a partial agonist at α4β2 and a full agonist at α7 neuronal nicotinic receptors) [26,27,28,29], or sodium oxybate (sodium salt of γ-hydroxybutyrate, targeting GABA_B_ receptors) [30]. Searching for new strategies and compounds for treating AUD is necessary [28,31]. Given the implication of peptides on alcohol-induced neurotransmission alteration, research focusing on the role of peptidergic neurotransmission molecular mechanisms in addictive disorders will offer new therapeutic possibilities [32].

This review analyzes recent research on the influence of the neuropeptide neurotensin (NT) in the neural circuitries altered by ethanol in the brain. In addition, we propose considering NT receptor (NTR) ligands as a possible pharmacological tool to apply in some forms, manifestations, and stages of AUD.

## 2. Neurotensin and NT Receptors

### 2.1. Neurotensin

The tridecapeptide NT was isolated from the bovine hypothalamus (Figure 2). It is involved in gut motility and belongs to a family of bioactive peptides, including contulakin, xenopsin, LANT-6, and neuromedin, showing similar amino acid sequences at the C-terminal region; this sequence is essential for the physiological effects mediated by these peptides [33].

NT arises from the pro-NT/neuromedin precursor (170 amino acids in humans). After its cleavage by prohormone convertases (PC, endopeptidases), the following molecules originated: large NT (140 amino acids), large neuromedin (125 amino acids), neuromedin (6 amino acids), and NT (13 amino acids). In addition, another fragment (24–140 amino acids) originated during the precursor processing; this fragment is stable in human serum and could be used as a biomarker for NT release [33]. NT is less stable than the large neuromedin in blood, and hence effective forms can exert longer-lasting biological effects due to its enhanced bloodstream stability compared to that reported for fully processed peptides [33]. It is important to note that, depending on the PC involved and expressed in the tissues (e.g., adrenal gland, intestine, brain), the peptides arising from the pro-NT/neuromedin precursor are different. Thus, PC1 originates large neuromedin and NT (intestinal pattern): PC2, neuromedin, and NT (brain pattern) and PC5A, NT, and neuromedin large forms (adrenal gland pattern). The pro-NT/neuromedin precursor is highly conserved in vertebrates. The coding region of the NT/neuromedin gene is spread over four exons and separated by three introns; it is transcribed to two transcripts (1.0 and 1.5 kb) differing in the 3′-untranslated region. In the brain, both mRNAs are equally expressed.

Experimental preclinical evidence supports the possible involvement of NT in many physiological and pathophysiological processes related to gut motility, bile acid release, glucose homeostasis, lipid metabolism, dopamine release, locomotor activity, blood pressure, angiogenesis, energy balance, body temperature, feeding, reproductive mechanisms, inflammatory processes, memory, stress, motivational and affective behaviors, antinociception, cancer, and alcohol intake [33,36,37,38,39,40,41,42,43,44]. Upregulation of NT and NTR1 has been demonstrated in patients with ulcerative colitis. The involvement of NT in irritable bowel syndrome has also been reported [37,45]. NT shows close anatomical and functional relationships with the mesocorticolimbic/nigrostriatal dopaminergic system, and it mediates some of the sensitizing and rewarding properties of drugs of abuse [46]. Moreover, NT seems to mediate the antidopaminergic action of drugs in preclinical studies, making NTR ligand candidates worth exploring for new treatments [33,47]. In this sense, NT agonists have been proposed to treat schizophrenia, drug addiction, and stress-related neuropathic pain. Neuromedin displays similar actions to those exerted by NT. Neuromedin and NT are co-released and co-localized; both peptides bind with a similar affinity to NTRs, but neuromedin is less potent on NTR1/NTR3 than NT [48,49]. Neuromedin is inactivated and degraded by aminopeptidases and NT by metalloendopeptidases [49].

### 2.2. Brain Distribution of Neurotensin

Pro-NT/neuromedin precursor mRNA expression has been reported in the caudate-putamen, hippocampus, accumbens nucleus, amygdala, bed nucleus of the stria terminalis, arcuate nucleus, paraventricular hypothalamic nucleus, medial preoptic area, cuneiform nucleus, periaqueductal gray, and dorsal raphe [33]. It has been demonstrated that steroid hormones and several neurotransmitters regulate the expression of the precursor in some of the previously mentioned regions of the central nervous system [33].

NT and its receptors are widely distributed within the mammalian central and peripheral nervous systems, and this widespread distribution confirms the many physiological actions related to the neurotensinergic system. For example, NT has been located in the bed nucleus of the stria terminalis, putamen, caudate, ventrolateral septum, subiculum, accumbens, globus pallidus, amygdaloid nuclei (e.g., central nucleus), lateral septum, thalamus (e.g., periventricular and intralaminar nuclei), hypothalamus (e.g., lateral and paraventricular nuclei), median eminence, midbrain (e.g., periaqueductal gray, midbrain limbic structures, substance nigra), pons-medulla oblongata (e.g., spinal and trigeminal substantia gelatinosa, area postrema, the nucleus of the solitary tract, nuclei parabrachialis medialis and lateralis, locus ceruleus, raphe nuclei, ventral tegmentum), and spinal cord. In some of the previous regions, NT has been involved in alcohol intake. For example, NT has been observed in the paraventricular nucleus of the thalamus, a nucleus associated with stress, motivation, and alcohol-related behaviors, and it has been suggested that in this nucleus, NT regulates alcohol/drug intake and reinstatement. Neurotensinergic neurons located in the central nucleus of the amygdala are activated after ethanol consumption, and a reduced ethanol consumption was observed after the genetic ablation of these neurons [50]. Concerning the reward circuits, NT and NTRs are mainly found in the striatum (involved in the progression from voluntary-driven to habit-driven automated and compulsive behaviors). Dopamine neurons associated with reward are primarily found in the VTA [51].

### 2.3. Neurotensin Receptors

NT binds to three different receptors: two G-protein-coupled receptors (class A), namely NTR1 and NTR2, and a single domain transmembrane protein, NTR3/Sortilin [52,53]. Class A GPCR, to which NTR1 and NTR2 belong, frequently forms homo and heterodimers with distinct properties affecting the binding affinities of ligands, signaling, and receptor endocytosis [54,55,56]. Functional heterodimers of dopamine (D2 and D3) and NTR1 receptors have been described, and the development of high-affinity bivalent ligands for these complexes opens new windows for the pharmacological control of their activity [55,56].

#### 2.3.1. NTR1

The human NTR1 is a seven-transmembrane domain protein (UniProt code P30989) [57] (Figure 3A) encoded by the *NTSR1* gene. The protein is made up of 418 amino acids. It exhibits typical G-protein-coupled receptors post-translational modifications, including glycosylation of asparagines 4, 37, and 41, a disulfide bridge between cysteines in positions 141 and 224, and two lipidation cysteine sites at positions 381 and 383 (Figure 3B). In the central nervous system, NTR1 regulates food intake and addiction neurotransmission. It also controls gastrointestinal and cardiovascular systems and behaves as a growth factor in different cells, affecting cell growth and survival [52,53].

Analysis of several crystal structures of NTR1 with X-ray diffraction or electron cryomicroscopy (cryo-EM) methodologies provided essential information regarding the dynamics of the receptor. Robertson et al. determined the inactive state of the receptor bound to a nanobody [61]. Furthermore, the structure of NTR1 attached to the carboxy-terminal region of NT (NT8–13) revealed that the hexapeptide binds almost perpendicularly to the plasma membrane plane, directing the carboxy end toward the center of the receptor protein (Figure 3D) [59,62]. The structural analysis by Kato et al. [63] disclosed conformational states (canonical and non-canonical) of NTR1 in complex with heterotrimeric Gi1 protein. Cryo-EM resolution of the NTR1 structure bound to NT and G protein Gαi1β1γ1 in a lipid environment showed a tight interaction receptor-G protein in a lipid bilayer that explains the mechanism of activation of G proteins and the activated intracellular signaling [64]. These structural conformations may have relevant functional significance and help understand the transition structures that the receptors display depending on their operational situation. Cryo-EM of human NTR1 disclosed that the phosphorylated receptor (on residues on the intracellular loop three, ICL3, and the C-terminal region) builds a stable complex with β-arrestin1; also, the membrane phospholipid phosphatidylinositol-4,5-bisphosphate (PIP2) bridges transmembrane domains 1 and 4 with β-arrestin [65].

Crystal structures of NTR1 bound to different ligands (partial agonists, inverse agonists, and full agonists) have revealed many receptor structures whose function depends on the type of ligand attached [59]. Inverse agonist SR48692 provokes the opening of the binding site by forcing the aperture of the extracellular portion of the transmembrane domains TM6 and TM7, making difficult the binding of the transducer G protein (Figure 3C). In contrast, full agonist peptide NT8–13 prompts the closure of the binding pocket (Figure 3D) [59].

Accurate determination of different receptor structures and resultant functional knowledge is essential to design specific and selective nonpeptide stable compounds that fine-tune and modulate the protein signaling according to the operational necessity.

#### 2.3.2. NTR2

The human NTR2 is a seven-transmembrane domain protein [58] (UniProt code P095665 [57]) (Figure 4A) encoded by the *NTSR2* gene. The protein consists of a sequence of 410 amino acids. It exhibits typical G-protein-coupled receptors post-translational modifications: a disulfide bridge between cysteines in positions 108 and 194 and a lipidation site at Cys377. No glycosylation targets appear on the ensemble (Figure 4B).

The NTR2 shows a lower affinity for NT than NTR1; it binds to the antihistaminic drug levocabastine, has low sensitivity for sodium ions, and presents no glycosylation sites on the extracellular N-terminal region [52]. NTR1 and NTR2 share approximately a 40% sequence identity as calculated by BLAST [66] and have a widespread but not identical tissue distribution [67]. The AlphaFold structure prediction shows that NTR2 outlines a seven-transmembrane serpentine structure similar to the neurokinin-1 receptor (Figure 4C).

#### 2.3.3. NTR3/Sortilin

The human NTR3/sortilin is a type I 100 kD glycoprotein receptor (Uniprot code Q-99523) [57] encoded by the *SORT1* gene and structurally unrelated to NTR1 and NTR2 with multifunctional tasks, including protein sorting and intracellular signaling, acting as a receptor, a co-receptor (heterodimerization), and a modulator of extracellular trafficking [68,69,70,71]. Two sortilin splice variants have been detected in human tissues (brain, heart, thyroid, or placenta, to name a few) [72]. The affinity of NT for the NTR3 stands in the low nanomolar range [52]. Because sortilin also inhabits intracellular vesicles, NT internalization stimulates its recruitment to the plasma membrane [69,72].

The protein has a signal peptide on the N-terminal end and a site for furine hydrolysis. It presents a single transmembrane domain and sequence analogies with Vps10p (on the extracellular domain) and CI-M6PR (cation-independent mannose 6-phosphate) receptor (cytoplasmic region) [52,72]. Figure 5 summarizes the complete amino acid sequence and the post-translational modifications of sortilin, as described (A). Panels B and C partially view the glycoprotein crystal structure bound to NT and resolved with X-ray diffraction.

The Protein Data Bank [35] holds a collection of structures of sortilin determined by X-ray diffraction. Figure 5C, represents the structure of the binding site of the C-terminal tetrapeptide of NT bound to the so-called ten-bladed-β-propeller domain of human sortilin. This tunnel structure also serves additional binding pockets for growth factors, thus evidencing the multifunctionality of this glycoprotein receptor [73].

### 2.4. Neurotensin Signaling Pathways

NTR1, NTR2, and NTR3 mediate the physiological effects mediated by NT through diverse intracellular signaling pathways (Figure 6).

After binding to NTR1, the peptide activates the Wnt/beta-catenin pathway and the transcription of the epidermal growth factor receptor and NTR1 genes; activates diacylglycerol lipase and phospholipase 2, promoting the release of arachidonic acid; favors cyclic adenosine monophosphate synthesis and activates protein kinase A, which regulates the transcription of the early growth response protein 1, activator protein 1, erythroblast transformation specific line-1 protein, and c-myc genes; activates focal adhesion kinase, proto-oncogen tyrosine-protein kinase, and small Rho GTPases, and the latter activates transcription factors. NT also activates phospholipase C-gamma, producing inositol triphosphate 3 and diacylglycerol. The former promotes the release of Ca^++^ from the endoplasmic reticulum, switching on protein kinase C, which, in turn, activates metalloproteinase and protein kinase D. Metalloproteinase cleaves and releases the epidermal growth factor that binds to its receptor, epidermal growth factor receptor 2 or epidermal growth factor receptor 3, thus favoring the activation of mitogen-activated protein kinase 1/2, mitogen-activated protein kinase, and RAF (rapidly accelerated fibrosarcoma). Epidermal growth factor-epidermal growth factor receptor binding activates the phosphatidylinositol 3-kinase-serine-threonine protein kinase pathway; NT favors the phosphorylation of insulin growth factor receptor 1 and triggers nuclear factor kappa light chain enhancer of activated B cells [38]. The signaling pathways mediated by NTR2/NTR3 are not well known; the main molecules involved in these pathways, after NTR2 or NTR3 activation, are mitogen-activated protein kinase 1/2 (for NTR2) and beta-catenin, tropomyosin receptor kinase B, ras homolog family member, ras-related C3 botulinum toxin substrate 1, protein kinase C, phosphatidylinositol 3-kinase, metalloproteinase, integrin, focal adhesion kinase, mitogen-activated protein kinase 1/2, epidermal growth factor receptor, and adenylyl cyclase (for NTR3) [38,74].

## 3. Neurotensin, Reward, and Alcohol

In this section, we review updated information on the relationship of NT with brain reward pathways and its implication on the effects provoked by ethanol on reward behavior. Reward implicates positive reinforcement, a process increasing the probability of occurrence of an earlier action followed by a favorable or pleasurable result, a crucial element for developing substance abuse by some individuals [75].

### 3.1. Brain Reward Circuits

Complex neuronal wiring modulated (activation and inhibition) by several neurotransmitters supports reward mechanisms within the brain. The midbrain’s ventral tegmental area node (VTA) houses afferent receiving (input) and efferent projecting (output) GABAergic, glutamatergic, peptidergic, and dopaminergic neurons projecting to and from different brain structures [15,21,75,76,77,78,79,80,81,82,83,84,85] (Figure 7).

Dopaminergic neurons in the VTA extend their axons to the NA (nucleus accumbens), prefrontal cortex (PFC), and amygdala (AMY), forming the mesocorticolimbic system, a critical network for reward management. A set of GABA interneurons within the VTA, modulated by μ-opioid receptors (μOR), play a significant role in reward mechanisms by inhibiting dopaminergic activity. One of the main destinations of VTA dopaminergic neurons’ output is the NA. In addition, the NA receives glutamatergic control inputs from the PFC, hippocampus, and basolateral amygdala (BLA) [25]. The PFC is responsible for executive control and planning to obtain the reward (for example, seeking a substance of abuse, including alcohol). Dopaminergic VTA and glutamatergic NA inputs inform this activity. The glutamatergic connections between the ventral hypothalamus (vHipp) and NA provide pertinent emotional information to modulate reward-related behavior.

The μ opioid receptors (μOP) localized in GABAergic interneurons within the VTA and other regions (rostromedial tegmental nucleus, RM, ventral pallidum, VP) are also crucial elements of reward dynamics. Activation of μOP abolishes the GABAergic inhibition of dopaminergic activity at the VTA. In addition, the direct activation of DA neurons by μOP may contribute to the positive reinforcement process (Figure 7) [75].

A midbrain-situated structure, the paraventricular nucleus of the thalamus (PVT) conveys distant and local signals modulating motivation, stress, affective behavior, and addiction. It connects through a large population of glutamatergic neurons with other brain nuclei (Figure 7) and also houses peptidergic neurons and may receive (putative) distant peptidergic information (substance P, neurotensin, neuropeptide Y, and orexin, among others) from other brain regions [21]. Further study of peptidergic pathways related to PVT controlling addictive behaviors is needed to ascertain better the mutual modulation of copious interconnected neurotransmitter signals [80,86].

The NA is the major component of the ventral striatum, relates to and modulates the limbic and motor systems, and contributes to the reward network [87,88]. Two structures with unique characteristics delineate the nucleus, the core, and the shell. It is a multiconnected center receiving and projecting neurons (releasing different neurotransmitters) to various brain regions. It receives, for instance, dopaminergic VTA projections, and it contains GABAergic interneurons and projecting neurons that connect with other brain centers (PFC, LH, PVT, vHipp) by glutamatergic, NT, and GABAergic neurons [88]. Dopamine 1 and 2 class receptors in NA regulate VTA intracranial self-stimulation (ICSS) [81].

AMY, or amygdaloid complex, is an important center in the brain networks of reward, emotion control, learning, and memory. Its connectivity secures its function and neuromodulatory performance. AMY contains several highly interconnecting nuclei (comprising the basolateral, cortical, central, and extended amygdala), projecting and receiving inputs from the PFC, NA, VTA, thalamus, and hypothalamus (Figure 7) [43,89]. Recent studies on in vivo mapping of the human AMY and neuroimaging studies provide new and applicable information that may help the setting of patient-directed modulatory therapies in brain behavioral alterations [90], including substance-abuse disorders [84].

The action of ethanol on reward and stress-management systems is complex and affects several neurotransmitter systems. The amount and the pattern of alcohol administration and individual genetic background influence dopaminergic, serotonergic, opioid, GABAergic, glutamatergic, and peptidergic neurotransmission and induce a mechanism of maladaptation that may recover rapidly or may take time, even if the ethanol exposure ceases [3,14,21]. A methodological drawback of animal experiments is the insufficient number of models of voluntary alcohol intake that permit neat extrapolation to humans. However, data extracted from several models and new experimental approaches (for example, ethanol vapor self-administration (ESAV) [91] are closing the gap. This point is relevant to determine better the brain regions and neurotransmitters involved in alcohol-induced damage.

After acute alcohol exposure, dopamine release in the NA and striatum is responsible mainly for the pleasurable effects experienced. Alcohol influences negative GABAergic control over dopaminergic neurons in the VTA and acts upon GABA activity’s presynaptic opioid regulation [75,83,91]. Additionally, alcohol affects the VTA’s glutamate control over GABA neurons projecting to VTA [92]. Consequently, dopamine release from VAT augments. Further alcohol administration decreases dopamine release in NA and striatum, and the circuitry demands more alcohol to obtain a reward. Other neurotransmitters, including peptides, modulate the transition, leading to maladaptation mechanisms. Stress control is a crucial element in conditioning brain responses. Persisting ingestion of alcohol over time may lead to a situation where alcohol withdrawal produces adverse effects and stimulates further consumption. Withdrawal relates to stress and anxiety-like responses, and the cycle of maladaptation consolidates. This stage links to the need to ingest more alcohol because the pleasurable effects are set at a higher cutoff point, and it is characteristic that dopaminergic and serotonergic activities decrease dramatically [82,93,94,95]. Peptidergic neurotransmission exerts a role in this stage, as corticotropin-releasing factors, neuropeptide Y, κ-opioid, substance P, NT, and other neurotransmission systems, take over stress responses and alcohol disturbances on neurotransmission homeostasis [15,19,20,82,96].

The participation of NT neurotransmission in reward circuits and its influence on dopamine release and action must be considered in the context of the functional complexity of dopaminergic neurotransmission in the NA and the regulation of dopamine release through DA2 autoreceptors modulating the function of dopamine transporters [97]. Distinct populations of MSN neurons (GABA medium spiny neurons expressing dopamine D1 and D2 receptors) resident in the NA, rich in DA1R and DA2R, may convey different outputs and responses to stimuli [98,99,100]. The chemogenetic/optogenetic studies of Strong et al. [101] showed that activation of DA1R neurons increased ethanol drinking in rats, and its inhibition reduced alcohol drinking. Additionally, stimulation of the DA2R subpopulation of neurons did not affect drinking behavior, but its inhibition augmented alcohol consumption. NT may act on DA neurotransmission and modulate reward responses in a not yet well-determined manner, probably dependent on localization and proximity to pre- and post-synaptic DA receptors. In this line, further research must clarify the mechanism by which NT affects DA activity and plays its role in the functioning of the reward network.

### 3.2. NT, Reward, and Stress

NT signaling in some areas, including AMY, lateral septum, BNST (bed nucleus of the stria terminalis), substantia nigra, or VTA (ventral tegmental area), is markedly abundant [102]. For a number of years, NT neurotransmission has been associated with reward mechanisms, substance abuse, and the pathophysiology of dementia and schizophrenia in the brain. The solid functional association of the NT grid with dopaminergic mesocorticolimbic and nigrostriatal systems makes NT participation in reward mechanisms a crucial element in understanding addiction [47,103]. NT inputs to the VTA have various origins: LH, NA, AMY, PFC, BNST, and periaqueductal gray matter (PAG). These multiple origins of NT neurons support their role in controlling motivational behavior [79,85] (Figure 7).

Evidence at the molecular level shows the interaction of NTR1 and dopaminergic type 2 receptors forming a heterodimer where NT modulates DA signaling, which strengthens the role of NT in reward in crucial brain areas concerned with reward mechanisms (see [46] for a review). The role of NTR1-biased activation of β-arrestin over Gq and its influence on dopamine 2 receptors (D2R) function (intracellular signaling, GABA release, and reward outcomes) [104] remains to be clarified.

NT signaling in the brain has been experimentally associated with stress and anxiety. For example, the infusion of NT or NT analogs into the PFC induced anxiogenic effects in rodents, and the administration of NTR1 inverse agonist SR48692 displayed no effects in normal rats but weakened the anxiety provoked by acute immobilization-provoked stress [105]. When studying chronic stress behavior in rats, Normandeau et al. [78] observed that the blockade of NTRs at the BNST eliminated stress-induced anxiety-like behavior. After challenging mice to chronic social defeat stress (CSDS), a group of male and female rodents named SUS (susceptible) avoided social interaction with same-sex, non-aggressive companions. By analyzing whole-cell recordings, the authors [43] identified a population of NT neurons in the lateral septum (LS) responding to stress triggered by social interaction only in SUS subjects but not in resilient (RES) or non-stressed control mice. Therefore, this NT group of neurons commands the processing of impeding social reward.

Furthermore, NT neurons from the LS projecting to the tuberal nucleus (TU) and the supramammillary nucleus (SUM) are responsible for hedonic feeding. Further studies will clarify the picture of NT signaling beyond the VTA, focusing on the connections and regulation of other regions (BNST, LH, central amygdala, CeA) or the parabrachial nucleus, PBN [51]). Therefore, the role of NT in reward goes beyond the VTA and involves other brain areas.

### 3.3. NT and Alcohol

The neurobiological background of AUD affects many neurotransmitters, including peptides, that modulate brain connections in various regions. This review focuses on NT, but it is essential to consider the participation of NT in a broader context, contemplating the action of other neurotransmitters [106].

Intracerebroventricular administration of NT in rodents enhanced ethanol-induced sleep and altered NT brain concentration (increased or decreased, depending on the brain structure) [107,108]. The injection of 5 μg of NT into the NA or PAG of rats potentiated ethanol (3 g/kg, intraperitoneally injected), induced hypothermia, and increased the sleep time provoked by ethanol by 50%. This effect was not observed when NT was administered directly into the caudate nucleus [109]. Additionally, chronic alcohol administration induced cross-tolerance to the inhibitory effects of centrally injected NT on locomotor activity [108].

Selectively bred mice for a different response to the hypnotic effect of alcohol, named long-sleep (LS) and short-sleep (SS), exhibited differences in the NTR densities (levocabastine-insensitive, NTR1, and levocabastine-sensitive, NTR2) in brain mesolimbic and cortical regions, and both acute and chronic alcohol administration downregulated neurokinin receptors 1 and 2 [110]. Lee et al. [40] reported that null NTR1 mice showed diminished sensitivity to the effects induced by ethanol on locomotion and consumed more alcohol when offered as a free choice versus water. Additionally, the NT analog, NT69L, augmented the sensitivity to alcohol in wild-type mice but did not affect the null NKR1 mice. The results suggested that NTR1 mediates NT-dependent alcohol intoxication. The same authors analyzed alcohol-induced ataxia, sleep, and hypothermia in a null mice model of NTR2. The study revealed that the null mice for NTR2 drank more ethanol and were less responsive to the ethanol acute sleeping effect than the wild-type progeny [111].

Long-Evans rats (selected because they consume more alcohol than other rat strains) had access to 20% ethanol for different periods with intermittent access to two-bottle-choice (IA2BC) [112]. The animals were grouped into high and low drinkers (showing different behavioral outcomes in various experimental settings). Heavy drinkers, not low drinkers, showed an NTR2 (protein) increase in the PVT. Moreover, administering NTR2 selective agonist JMV-431 to low drinkers mimicked the behavioral response caused by ethanol in heavy drinkers [112].

The amygdala’s central nucleus (CeA) contains a group of NT neurons projecting to the parabrachial nucleus (PBN). In vivo, exposure to low alcohol concentrations activated NT neurons within the CeA, and selective lesion of these neurons curtailed alcohol consumption and preference [50]. Optogenetic stimulation of CeA resident NT neurons projecting to the PBN augmented alcohol and tasty fluids intake. Therefore, the CeA-PBN NT connection seems to participate in the adaptive mechanisms induced by ethanol in the neural circuits linked to reward. Future research will clarify its importance and the influence of other neurotransmitters in this connection.

The association of NTR1 with dopamine 2 receptors (DA2R) leads to a reduced affinity of DA2R for dopamine in striatopallidal MSNs within the NA and cuts alcohol consumption. However, NT also promotes dopamine signaling in the dorsal striatum. NT may influence the effects of ethanol differently according to the brain region and the dose and patterns of consumption of the toxic substance [51]. In addition, DA2R heterodimerization with NTR1 and other GPCRs may impact the activity of dorsal and ventral striatopallidal GABA neurons through biased signaling [113,114]. To date, no clear evidence indicates DA1R heterodimerization with functional impact.

Consequently, how specifically NT influences DA1R and DA2R signaling requires further analysis. Additionally, NT regulates the glutamate corticostriatal limbic network and interacts with NMDA (N-methyl-D-aspartate) receptors [115], making NT a versatile modulator of reward and controller of the award disruptions caused by ethanol [116]. The balance of all the effects at present favors that activation of neurotensinergic systems generally leads to reinforced alcohol responses (narcosis, hypothermia, locomotor activity) with a consequent reduction of ethanol drinking by a combined regulation of dopamine and glutamate brain tones [103]. As indicated, differential effects after activation of DA1R and DA2R may explain the regulation of alcohol-drinking behavior and habit consolidation by NTR1 [101].

No strong genetic evidence linked to NTRs or NT has been observed to be associated with alcohol-drinking behavior or AUD development [117]. Of the hundreds of variants described for the NTR1, NTR2, and sortilin proteins in humans, no clear findings reveal their influence on alcohol-drinking behavior or damage [57]. Only a few studies have indicated a relationship between NTR1 variants and alcohol addiction and other brain diseases, such as schizophrenia. For example, the genotyping of SNPs (single nucleotide polymorphisms), rs6011914C/G, and rs2427422A/G in NTR1 in a male Han Chinese population determined an association of the SNPs with alcohol dependence [39]. Analysis of QTLs (quantitative trait loci) in mice and rats defined that the genes regulating NTR1 and NT expression in several brain areas may be the same as the genes modulating ethanol sensitivity (estimated hypothermia, sleep, and altered locomotion activity) [118,119,120].

Because of the difficulties in developing and advancing clinical research (heterogeneity of AUD, genetic backgrounds, patterns of ingestion, or limitations of experimental procedures in humans), we do not have specific data suggesting a precise role for NT in the pathogenesis of AUD. Therefore, the analysis of the implication of NT in the alterations generated by ethanol after acute and chronic exposure needs further and deeper experimental exploration.

## 4. NT Receptors Ligands as Possible Pharmacological Treatments of AUD

G-protein coupled receptors (GPCRs), in general, do not only behave as two distinct on-and-off activation states but also exhibit activity independent of ligands that may be modulated allosterically. Additionally, they frequently show biased responses that amplify their meta states and functional role. The reaction of GPCRs to ligands with agonist, antagonist, inverse agonist, or allosteric properties and the binding to transducers makes these receptors susceptible to regulation in a complex and versatile fashion [121,122,123,124,125,126,127]. Consequently, drugs designed to control and modulate GPCRs have enormous potential.

Some characteristics of NT ligands and their influence on the receptor ensemble conformation leading to the activation of intracellular signaling pathways are briefly described. Different drugs selectively bind to the NTR1 (either the orthostatic or allosteric sites), affecting the dynamics and function of the receptor ensemble in singular manners (full agonists, partial agonists, inverse agonists, allosteric modulators, and bivalent ligands) (see representative structures in Figure 8).

Combining available crystallography studies, quantum calculation, and NMR techniques, Bumbak et al. [123] provided motion states of NTR1 under the μs scale that reveal a close correlation between pharmacological effects induced by ligands and conformer structures of the receptor. In addition, the NTRs may acquire several conformational states with functional significance depending on the ligand they bind and the transducers they couple to (G proteins and β-arrestins).

The hexapeptide NT8–13 full agonist includes the carboxyl end’s last six amino acid residues of NT. It exhibits similar biological activity compared to NT and represents the necessary structure fitting to the orthostatic receptor site through specific interactions through the aromatic ring of Tyr11 and the basic-lengthy side chain of Arg9 bearing a guanidinium group, among others [128,129]. It binds to NTR1 and contracts the receptor structure by approaching transmembrane TM6 and TM7 helices and giving access to a G-protein site on the cytosolic domain [59] (see Figure 3D). Several peptidases rapidly hydrolyze this peptide, and efforts to stabilize the molecule and obtain peptides with agonist capacity and resistance to degradation include the methylation, acylation, and substitution of Ile12 for tert-butyl glycine [130].

The efforts to synthesize small-size nonpeptide ligands that are stable, safe, and selective for NTR1 have produced several compounds with different binding capacities and functional effects. SRI-9829 is a nonpeptide indole-based structure with full agonist capacity (Figure 8) [59]. The substitution of the adamantyl amino acid moiety of compound SR-48692 (an NTR1 antagonist) with L-Leu generated a compound, RTI-3a, with partial agonist activity [131] (Figure 8).

SR-48692 and SR-142948A drugs display different activity spectra on NTRs [132,133]. However, both inhibit basal activity and antagonize NTR1-dependent activity (inverse agonism) [134]. Inverse agonists, contrary to full agonists, expand the TM6 and TM7 domains of NTR1 and impede access to the G-protein binding cytosolic site [59] (see Figure 3C).

NTR1-biased ligands (β-arrestin-biased allosteric modulators, BAM) (Figure 8) offer the advantage of triggering one specific transducer over another (heterotrimeric G-protein and β-arrestin), thus impacting one signaling route alone and, in some cases, inactivating the alternative route that may lead to undesired effects. The nonpeptide quinazoline derivative, ML-314, exhibits agonist-allosteric and biased features that make it suitable to fine-tune the receptor protein toward specific intracellular biochemical events. The compound favors the formation of NTR1/β-arrestin complexes and allosterically regulates the binding of NT to NTR1 [135,136]. Data from the determination of 13Cε-methionine chemical shift-based global order parameters indicate that ML-314 changes the conformation of the receptor protein and favors an even better-suited interaction of the agonist NT8–13 to the orthostatic binding site [123].

Another nonpeptide quinazoline derivative, SBI-553, presents agonist and allosteric properties biasing the function of NTR1 by activating the coupling of the receptor to β-arrestin and simultaneously blocking the pairing of a heterotrimeric G-protein, leading to specific intracellular signaling that depends on β-arrestin action only. In this manner, a single compound modulates and directs a particular receptor action, whereas other signaling mechanisms remain silenced [137,138]. In NK1R-transfected HEK293T and U2OS cells, SBI-553 activated receptor phosphorylation, recruitment of β-arrestin, and receptor internalization. This compound also enhanced the agonist effect of low NT concentration on receptor phosphorylation, recruitment of β-arrestin, and receptor internalization. The impact of SBI-553 is allosteric be cause the specific pharmacological antagonism of NTR1 blocked NTR activation but did not influence SBI-553’s outcomes. While NT stimulated inositol 1,4,5-trisphosphate formation and calcium mobilization through activating Gq protein, SBI-553 did not affect Gq activation [138]. Additionally, the drug offers better pharmacokinetic properties than ML314 [137].

A group of ligands that show bivalent binding capacity may serve as valuable tools to target heterodimeric receptors. Bivalent ligands can bind two different proteins and hence regulate their activity. In their core structure, two pharmacophores are attached through a linker structure. Figure 8 shows the construction of two hetero-bivalent compounds that bind to the dopamine receptor 2-NTR1 complexes (heterodimers with a significant function in addiction and other brain disorders). Interestingly, the length of the linker moiety determines the affinity for different heterodimers where NTR1 participates [55,56]. Moreover, homo-bivalent ligands for the dopamine 2 receptor promote their homodimerization and reduce the formation of dopamine 2 receptors-NTR1 heterodimers, thus influencing the activity of homo and heterodimers [55]. Therefore, synthesizing bivalent ligands opens a new window for drug design directed to the modulation of NTR1 activity in alcohol addiction and other pathologies.

In general, the relevance of NT ligands in AUD lies in restoring dopamine-signaling homeostasis disrupted by alcohol or other abused substances. However, we need to extend the study of the multiple roles that NT may play in reward mechanisms and anxiety behaviors linked to the risk of alcohol addiction. With clear information, we might be able to apply adequate NT ligands displaying agonistic, antagonistic, bivalent, or biased agonistic-allosteric properties to specific AUD clinical entities.

Table 1 summarizes a few examples of preclinical experimental approaches with NT ligands that may open new therapeutic strategies in AUD.

## 5. Conclusions and Future Perspectives

Basic experimental research in rodents evidences the participation of NT in the control of reward and stress-related brain circuits. However, the precise modulatory role of this neuropeptide is not fully understood. Unraveling NT involvement in disrupting reward homeostasis by ethanol and other addictive substances deserves attention. We need to define neural circuits, determining molecular events where NT activation/deactivation influences other signaling mechanisms (by cross-talking with other neurotransmitter intracellular signals or by modulating the binding to other protein receptors to which NTRs heterodimerize and modify their function). Mainly, how NTR1 interacts with DA1R and DA2R signaling deserves attention to understand the reward system better because both dopamine receptors seem to behave in an opposed fashion concerning pleasurable experiences and habit consolidation. Developing stable, selective, and non-toxic NT analogs that easily reach the brain is paramount to controlling NT intracellular action. In addition, selecting appropriate animal models (inbred strains, animal models of “voluntary” access to alcohol) will shed light on the influence of NT on alcohol damage.

Careful and well-designed clinical trials with safe NT ligands may provide new insights as to whether NT signaling may prove a target for the pharmacological treatment of some forms and manifestations of AUD in selected patients.

Regulation of dopamine signaling through the control exerted by NTR1 may justify using NT ligands to pharmacologically restore dopamine homeostatic action in the brain when it is disturbed by substance abuse, including harmful alcohol consumption. One drawback, however, is that NTR1 controls other cellular outputs that influence vital functions such as body temperature, motor control, mood modulation, or blood pressure maintenance. The preference of biased agonist-allosteric compounds for a signaling route over another opens new possibilities for drug design directed at selecting therapeutic consequences and avoiding non-desired secondary effects managed by the same protein. This property may provide new valuable pharmacological tools driving a specific and searched-for result. Biased drugs offer a unique landscape of opportunity that should be explored in basic research and clinical assays.

Given the functional versatility and tissue distribution of NTR1, drug design focused on NTR1 may result in safe and specific compounds with therapeutic potential in a broad collection of pathologies with high incidence, not only in substance abuse but also in diabetes, obesity, schizophrenia, or cancer [46,55,56,123,137,138,141,142]. Recent detailed structural and dynamic studies of the NTR1 [56,59,122,123] provide helpful information for designing compounds that provoke specific receptors’ functional states affecting several pathologies.

Here we present a broad landscape of NT’s functional role considering anatomical, biochemical, physiopathological, and drug design attributes to understand better its participation in human pathologies awaiting secure and effective pharmacological therapies. Undoubtedly, alcohol use disorders represent a field where research on NT modulation of brain neurotransmission may generate satisfactory results.

The analysis of the participation of NT in the physiopathology of AUD needs further and deeper exploration, both in animal models and cell cultures, in vitro studies of molecular mechanisms, as well as in humans (considering the apparent restrictions) to ascertain its role and open the possibility of using NT analogs to pharmacologically target the receptors to modulate and reverse some symptoms and clinical manifestations of AUD.

## Figures and Tables

**Figure 1 ijms-24-08656-f001:**
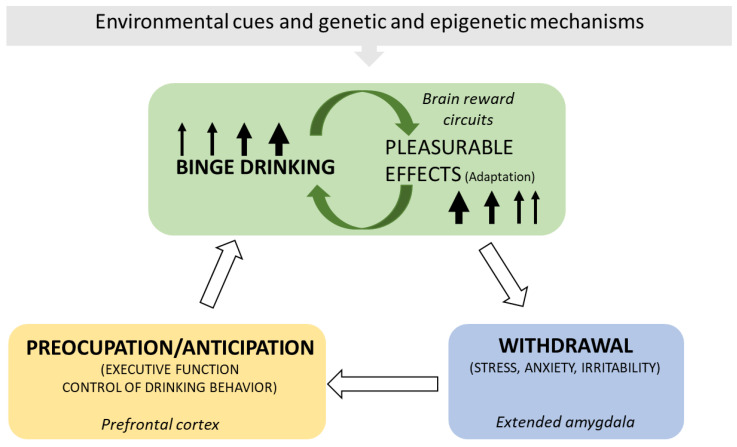
The brain cycle of AUD. The scheme depicts the principal elements and consequences of the brain’s maladaptation to ethanol ingestion, including the chief brain regions implicated. Binge alcohol drinking stimulates the brain’s reward circuitries and generates pleasurable effects. Pleasure conditions the demand for more alcohol. With time, the pleasure experience diminishes, and alcohol drinking escalates to obtain the previously experienced result (tolerance). Alcohol withdrawal produces discomfort, anxiety, and stress, which disappear with alcohol ingestion. Alcohol alleviates the unpleasant and negative emotional experiences derived from the absence of ingestion but no longer produces pleasurable effects. Relapses ease withdrawal manifestations, and the maladaptation circle keeps going. Genetic and epigenetic mechanisms and environmental cues contribute to this brain maladaptation cycle [9,10,11,12,13,14,15].

**Figure 2 ijms-24-08656-f002:**
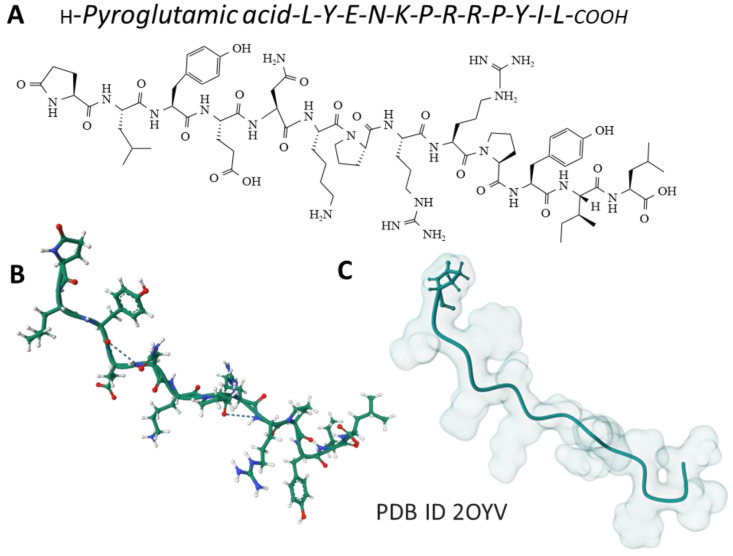
The amino acid sequence of NT (**A**) and its structure determined by NMR spectroscopy in a membrane-mimicking environment (**B**) [34] is from the Protein Data Bank [35]. The Gaussian volume of the tridecapeptide depicts its 3D architecture (**C**).

**Figure 3 ijms-24-08656-f003:**
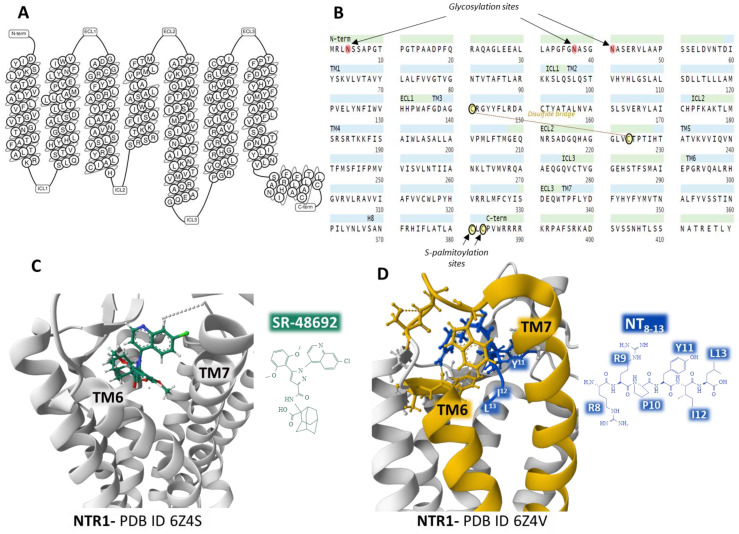
Structure of the seven transmembrane domains of the NTR1 (**A**) [58] and complete amino acid sequence of NTR1, indicating post-translational modifications (**B**) [57]. Structure of the NTR1 bound to the inverse agonist SR-48692 and the position of transmembrane domains TM6 and TM7 (**C**) [59]. Installation of the agonist peptide NT_8–13_ bound to NTR1 (**D**) [59]. (**C**,**D**) were obtained from the Protein Data Bank [35] and modified with the free-web-based software Mol* (https://molstar.org/ (accessed on 4 March 2023)) [60].

**Figure 4 ijms-24-08656-f004:**
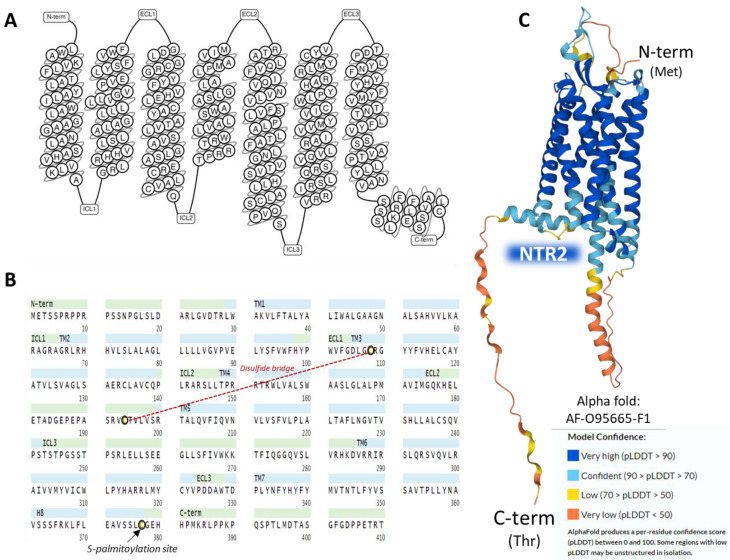
Structure of the seven transmembrane domains of NTR2 (**A**) [58]. The complete amino acid sequence of NTR2 indicates post-translational modifications (**B**) [57]. The AlphaFold prediction of the structure of NTR2 is represented in (**C**) [57].

**Figure 5 ijms-24-08656-f005:**
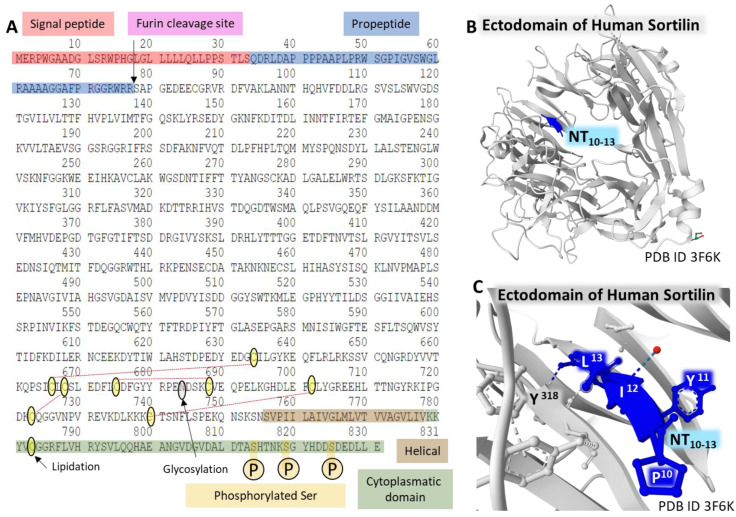
Structure of NTR3/Sortilin. The complete amino acid sequence of the receptor (**A**). The figure highlights some domains of the protein and post-translational modifications (the furin site that cleaves the propeptide; lipidation of Cys^783^; glycosylation of Asn^684^; phosphorylation of Ser^814^, Ser^819^, and Ser^825^; and disulfide bridges between Cys^634^-Cys^666^, Cys^668^-Cys^723^, Cys^675^-Cys^688^, and Cys^702^-Cys^740^) [57]. Structure of the C-terminal tetrapeptide of NT bound to the ectodomain of human Sortilin (**B**,**C**). These structures corresponding to the ectodomain of human sortilin bound to an NT fragment [73] are from the Protein Data Bank [35], drawn with the free-web-based software Mol* (https://molstar.org/ (accessed on 4 March 2023)) [60].

**Figure 6 ijms-24-08656-f006:**
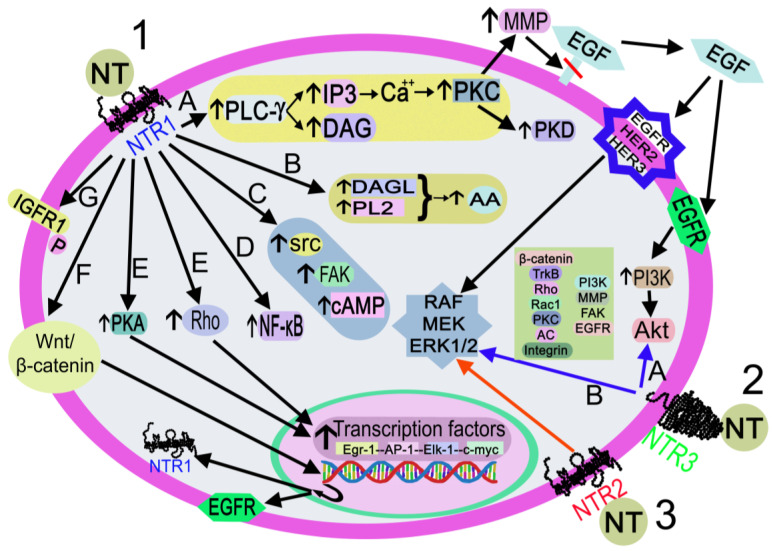
Activation of the signaling pathways after the binding of NT to NTR1 (black arrows), NTR2 (red arrow), or NTR3 (blue arrows). **1**. NT-NTR1 binding. A: DAG/I3P synthesis after PLC-γ activation. IP3 promotes the release of Ca^++^, which activates PLC, PKD, and MMP. MMP cleaves and releases EGF. EGF-EGFR binding activates the PI3K-Akt pathway, and EGF-EGFR/HER2/HER3 binding activates RAF/MEK/ERK1/2. B: DAGL/PL2 activation promotes the synthesis of AA C: FAK/src/cAMP activation. D. NF-κB activation. E: Rho/PKA activation interacts with transcription factors (Egr-1, AP-1, Elk-1, c-myc). F: activation of the Wnt/beta-catenin pathway promotes the EGFR/NTR1 gene transcription. G. IGFR1 phosphorylation. **2**. NT-NTR3 binding. A: Pathways involve beta-catenin, TrkB, Rho, Rac1, PKC, AC, integrin, PI3K, FAK, and EGFR B: MAPK activation. **3**. NT-NTR2 binding triggers MAPK activation. Abbreviations: AA: arachidonic acid; AC, adenylyl cyclase; Akt, serine-threonine protein kinase; AP-1, activator protein 1; Ca^++^, calcium cation; cAMP, cyclic adenosine monophosphate; c-myc, proto-oncogene protein; DAG, diacylglycerol; DAGL, diacylglycerol lipase; EGF, epidermal growth factor; EGFR, epidermal growth factor receptor; Egr-1, early growth response protein 1; Elk-1, erythroblast transformation specific line-1 protein; ERK1/2, mitogen-activated protein kinase 1/2; FAK, focal adhesion kinase; HER2, epidermal growth factor receptor 2; HER3, epidermal growth factor receptor 3; I3P, inositol triphosphate 3; IGFR1, insulin growth factor receptor 1; MEK: mitogen-activated protein kinase kinase; MMP, metalloproteinase; NF-κB, nuclear factor kappa light chain enhancer of activated B cells; NT, neurotensin; NTR1, neurotensin receptor 1; NTR2, neurotensin receptor 2; NTR3, neurotensin receptor 3; P, phosphorylation; PI3K, phosphatidylinositol 3-kinase; PKA, protein kinase A; PKC, protein kinase C; PKD, protein kinase D; PL2, phospholipase 2; PLC-γ, phospholipase C-gamma; Rac 1, ras-related C3 botulinum toxin substrate 1; RAF, a serine/threonine-specific protein kinases (rapidly accelerated fibrosarcoma); Rho, small Rho GTPases; src, proto-oncogen tyrosine-protein kinase; TrkB, tropomyosin receptor kinase B; Wnt/β-catenin, Wnt/β-catenin pathway.

**Figure 7 ijms-24-08656-f007:**
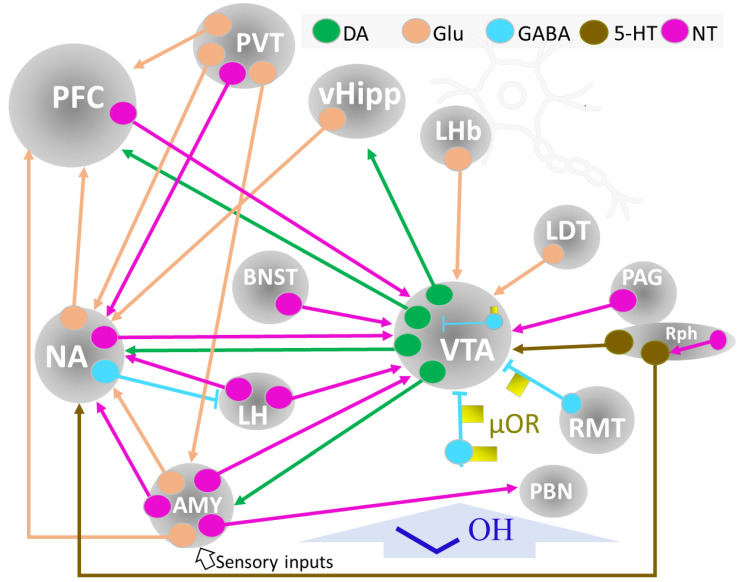
Summary of the circuits and brain structures implicated in reward and motivation. Ethyl alcohol (dark blue) interferes with neurotransmission within this circuitry and induces maladaptation processes depending on dose, pattern, exposition time, and individual responses. Abbreviations: Neurons: DA, dopaminergic; Glu, glutamatergic; GABA, gabaergic, 5-HT, 5-hydroxytryptaminergic; μOR, mu-opioid receptor; NT, neurotensinergic. Brain structures: AMY, amygdala; BNST, bed nucleus of the stria terminalis; LDT, lateral dorsal tegmentum nucleus; LH, lateral hypothalamus; LHb, lateral habenula; NA, nucleus accumbens; PAG, periaqueductal gray matter; PBN, parabrachial nucleus; PFC., prefrontal cortex; PVT, paraventricular nucleus of the thalamus; RMT, rostromedial tegmental nucleus; Rph, Raphe; vHipp, ventral hippocampus; VTA, ventral tegmental area.

**Figure 8 ijms-24-08656-f008:**
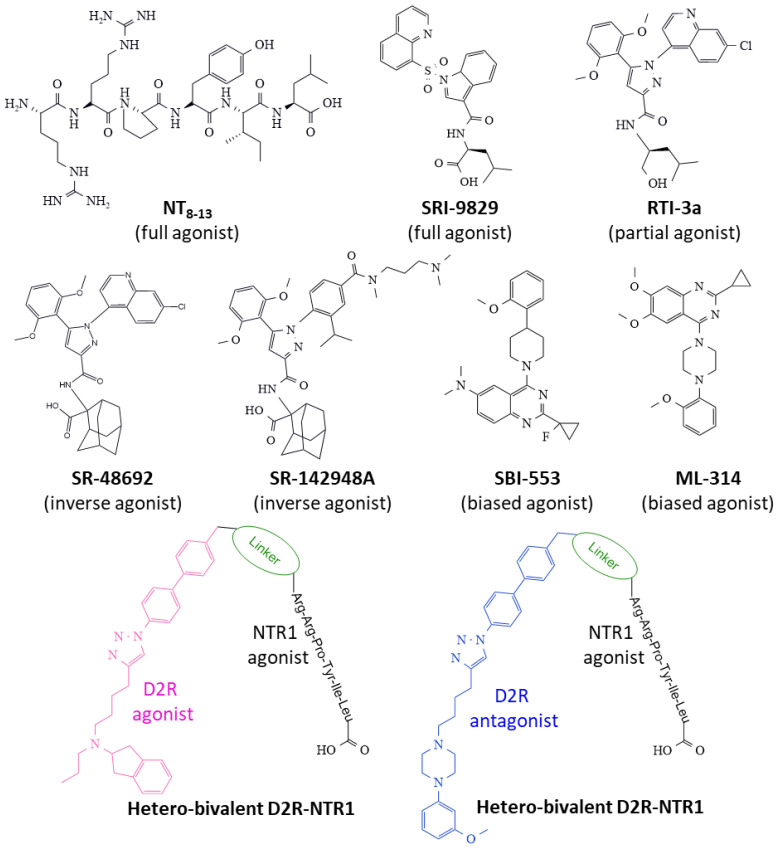
Representative structures of NTR1 ligands binding to the receptor can expand (inverse agonists) or contract (full and partial agonists) the orthostatic receptor pocket, inciting distinct structural conformers that recruit transducers and trigger differentiated intracellular biochemical events. β-arrestin-biased allosteric agonists (B.A.M.) induce a specific β-arrestin response, avoiding Gq activation. The figure also depicts two hetero-bivalent ligands targeting Dopamine 2 receptors-NTR1 heterodimers. Abbreviations: D2R, dopamine 2 receptors.

**Table 1 ijms-24-08656-t001:** Representative in vitro and in vivo experimental models on compounds modifying NT activity. Abbreviations: CHO, Chinese hamster ovary cells; COS, fibroblast-like cells isolated from monkey kidney; HEK, human embryonic kidney cells; NT, neurotensin; PI, phosphatidylinositols; U2OS, human bone osteosarcoma epithelial cells.

**Compounds**	**Experimental Model**	**In Vitro Studies**	**References**
Hetero bivalents DR2-NTR1 (agonist/antagonist properties)	Cultured CHO cells expressing receptors	Radioligand saturation and BRET assays to determine affinities and promotion of D2R homodimerization and heterodimerization.	[55,56]
NT_8–13_ analogs	Cultured CHO-K1 cells expressing rat and human NTR	Radioligand saturation analysis for determination of crucial amino acid positions related to receptor affinity and PI turnover and stability.	[128,129,130]
Small non-peptide ligands of NTR: SRI-9829, RTI-3a, SR-48692, SR142948A	Transfected cultured HEK293, COS-7 cultured cells	Structural, ligand binding, and signaling assays.	[59]
NT_8–13_ and SR142948A	Transfected cultured HEK293 cells	Structural dynamics of NTR1 with analysis of its global motions.	[123]
Derivatives of SR-48692 antagonist	CHO-K1 cell line expressing NTR1	Calcium mobilization assays	[131]
SR-48692 antagonist	Cultured transfected COS-7 cells, and brain tissue, form different species,	Radioligand binding assays	[132,133]
Nonpeptide ML314 biased agonist of NTR1	β-arrestin conjugated (GFP) reporter expressed in a U2OS cell line	Activates the β-arrestin pathway and blocks G-protein -dependent signaling.	[135,136]
Quinazoline NTR1 modulator, SBI-553	In vitro radioligand binding, NTR1-β-arrestin-GFP, and calcium flux assays in HEK-293 transfected cells	Activates β-arrestin signaling and inhibits Gq-protein-dependent calcium signaling.	[137]
Quinazoline NTR1 modulator, SBI-553	In vitro assays in HEK 293 and U2OS transfected cells	Promotes receptor phosphorylation, receptor internalization, and β-arrestin activation.	[138]
**Compounds**	**Experimental Model**	**In Vivo Studies**	**References**
Nonpeptide ML314 biased agonist of NTR1	Injection in rats	Inhibits amphetamine self-administration.	[136]
Silencing of lateral septum neurotensin-positive neurons	In vivo suppression of NT neurons in LS in C57BL/6J mice	Promotion of palatable feeding.	[139]
In vivo genetic ablation septum neurotensin-positive neurons (LS^NT^)	Male and female C57BL/6J mice	LS^NT^ modulates social interaction and social reward (stress susceptibility).	[140]
NTR1 agonist PD- 149,163, and antagonist SR-48692	Brain microinjections in male Sprague–Dawley rats	PD-149163 induces anxiogenic effects when injected into the prelimbic region of the medial prefrontal cortex.	[105]
Nonpeptide ML314 biased agonist of NTR1	Intraperitoneal injection in C57BL/6J mice	Attenuates methamphetamine-induced hyperlocomotion and methamphetamine-associated conditioned place preference.	[136]
Quinazoline NTR1 modulator, SBI-553	Intraperitoneal injection and oral administration to DAT KO C57BL/6J mice (dopamine-depleted dopamine transporter knockout)	Attenuates hyperdopaminergic activity.	[137]
Quinazoline NTR1 modulator, SBI-55	Intraperitoneal injection in cocaine self-administration in C57BL/6J mice	Attenuates cocaine-induced hyperlocomotion without producing non-desired side effects dependent on Gq protein activation.	[138]
Selective NTR2 agonist JMV-431	Brain injection in rats consuming alcohol (chronic consumption)	Chronic and excessive alcohol consumption induces behavioral changes, partly mediated by the NTR2.	[112]

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
