# Peer review of "Neurotensin and Alcohol Use Disorders: Towards a Pharmacological Treatment"

_ijms, 2023, doi:10.3390/ijms24108656_

Round 1
Reviewer 1 Report
The review by Rodrigues F.D.et al is focused on the role of neurotensin (NT) in alcohol addiction and alcohol use disorders (AUD). A broad picture of the structure of NTs and its receptors NTR1, NTR2, NTR3 (sortilins) and their functional role in regulating the brain reward circuits associated with alcohol addiction is presented. The material of the review is quite difficult to understand therefore, a good find of the authors is a visual schematic representation of the signaling pathways activated after NT binding to NTRs (Fig.6) where numbers 1, 2, 3 indicate the pathways related to a particular receptor (NTR1, NTR2 and NTR3, correspondingly). Also the next Fig.7 well and clearly presents the circuits and brain structures associated with reward and motivation. It is also successful to mark signals from dopaminergic neurons (DA) by one color, glutaminergic (GA) neurons by other color etc. At the same time Fig.7 will be easier to understand if abbreviations are placed not alphabeticaly but in semantic order. I’d recommend such an order: Abbreviations: neurons (DA)-dopaminergic, (GA)- glutaminergic, (GABA) - g-aminobutyric interneurons …, ; µOR - µ-opioid receptor; Brain structures: (AMY) amygdala, … Out of respect for the reader, it would be more polite to use common words more often, rather than special terms. For example, terms afferent and efferent might be changed at least sometimes for input and output.
The final section of the review “NTR ligands as possible pharmacological treatment of AUG” must be updated by a Table with several most successful examples of such treatment on animal models and columns marked “model, mechanism, reference”.
There are also some technical disorders spoiling the corresponding figures because some right marking numbers are shifted to the left: 438-457 (Fig.7) and 663-686 (Fig.8).
In general, the review deserves the highest praise. It contains a lot of new information, outlines clear ways of pharmacological treatment, will be very useful for practicing physicians and must be published in IJMS.
Out of respect for the reader, it would be more polite to use common words more often, rather than special terms. For example, terms afferent and efferent might be changed at least sometimes for input and output.
Author Response
Thank you for taking the time and expertise to review our manuscript and your valuable suggestions. We have introduced changes in the MS according to the comments made as follows:
-We have changed the order of the abbreviations in the legend of Figure 7 so that different types of neurons and brain structures appear now separated (lines 615-624).
-The terms afferent and efferent have been changed in some paragraphs to input and output signals, as suggested (line 583).
-A new table (Table 1) compiles a few examples showing experimental evidence that supports the search for pharmacological strategies to treat AUD and other diseases associated with NT signaling (see page 19, from line 1030).
-The position of figures 7 and 8 in the text has been checked to avoid the disorder of lines pointed.

Reviewer 2 Report
The authors produced a very comprehensive review of current knowledge about neurotensin and how this peptide could be involved in alcohol use disorder.
This review encopasses a large spectrum.
1. a general review on alcohol use disorder pathophysiology,
2. a general overview of the history of neurotensin discovery and identification among other hypothalamus-produced neuropeptides,
3. a description of neurotensin production in several areas of the brain,
4. a complete description of the 3 known types of neurotensin receptors (including the genes coding for them) and their brain localisation
5. a comprehensive description of NT synthesis in neurons and neurotensin receptors' localization in key points of the brain reward circuit supporting the role of neurotensin in addiction acquisition, especially regarding the acute and chronic effect of alcohol
6. a specific paragraph on the putative role of neurotensin in the stress information processign and how it could be involved in stress-induced relapse in preclinical models of alcohol dependence
7. the potential role of NT ligands as pharmacological treatment of alcohol use disorder.
The review is very rich, well organised and documented, and I learned a lot while reading it. It reflects the high amount of work that was produced by the authors.
This work is worth publishing in the IJMS. I would only suggest minor modifications, all in the sense of reducing a tendency to overgeneralization of preclinical research results to Human pathology or treatment. I would suggest to choose a more cautious langage.
ABSTRACT:
the authors write that "Experimental evidence supports the participation of neurotensin (NT) in alcohol addiction." I would suggest to rephrase to "NT in preclinical models of alcohol addiction", as no experimental evidence supports the participation of NT in alcohol addiction in Humans.
the last sentence:
"the possible use of nonpeptide ligands modulating the activity of the NT system to alleviate or suppress harmful drinking behaviors leading to alcohol addiction and health ruin" could be rephrased in ".... alleviate or supress preclinical models of harmful drinking behaviors mimicing alcohol addiction leading to health ruin in Human beings."
INTRODUCTION
L54 "Around 56% of deaths were premature (before age 75) [7]". the authors could add that most of the health burden related to excessive alcohol use, and especially deaths, is supported by subjects with not only excessive alcohol use but severe AUD, formelly knwon as alcohol dependence (see: John, U.; Rumpf, H.; Hanke, M.; Meyer, C. Severity of alcohol dependence and mortality after 20 years in an adult general population sample. Int. J. Methods Psychiatr. Res. 2022, 21, e1915.) This fact support the need for specific treatment helping patients with AUD to stop or reduce their alcohol use.
The small paragraph "Brain peptidergic system ...viscious addiction circle" L102 to 108 would be more easy to read at the end of the previous paragraph L73, before the paragraph L 95 that introduces current available treatments.
When citing "Current approved pharmacological treatments..." L119, the authors could cite sodium oxybate (approved in Italy and Austria) or Topiramate (several interesting studies, even more than gabapentin).
NT AND NT receptors
The sentence L187 "NT has been involved in many physiological and pathophysiological processes such as gastric regulation, gut motility, bile acid release, glucose homeostasis, lipid metabolism, dopamine release, locomotor activity, blood pressure, angiogenesis, energy balance, body temperature, feeding, reproductive mechanisms, inflammatory processes, memory, stress, motivational and affective behaviors, antinociception, Alzheimer's disease, cancer, and alcohol intake [30,33-41]" should also be rephrased. The references cited are either reviews or book chapters, or preclinical studies. A Human study (36) is only a candidate gene association study comparing 127 AD patients and 131 healthy control, which is far from a definitve evidence of invovlment in Human physiopathology.
to that respect, the preclinical evidence is much more convincing and should be cited as it is.
L197: "Moreover, NT exerts an antipsychotic effect playing an essential role in stress-induced analgesia and psychostimulant responses [30,44]"
once again, neither ref 33 nor 44 are RCT of neurotensin in Human patients suffering from schizophrenia. It would be more correct to state that neurotensin seems to mediate the antidopaminergic effect of pharmacological agents in preclinical models, suggesting that neurotensin receptors ligands could be promising new treatments.
But I am not aware of a clinical developement of such a drug. Of course, drugs that are composed of peptides are difficult to deliver because they are digested when taken orally, but this could be discussed, as several new ways of delivery are more and more available.
Maybe this could be part of the last perspective paragraph : pros and cons the developpement of NT or NT-Receptors ligands as new pharmacological treatments of AUD.
Brain distribution of NT:
a sentence could summarize the findings of this paragraph saying that regarding the reward circuit, NT and NT R are not located in the VTA, were DA neurons orginate, but are located in the striatum (caudate, putamen and nuccleus accumbens), structures involved in the transition from volontary driven to habit-driven automated and compulsive behaviors).
NT Receptors:
this paragraph and the figure 6 is very useful. I learned a lot.
Brain Reward circuit:
This paragraph is interesting but some key elements of complexity are missing, especially the difference in D1 vs D2 dopamine tracks (see PMID: 33963288, or PMID: 36589290) in the striatum or the existence of D2 autoreceptors that regulates DA release (PMID: 36170827). Thus, the effect of NT or NT R ligands could exert a reverse effect on dopamine release if they are close to post- or pre-synaptic dopaminergic receptors.
this should be discussed as a limitation to our current knowledge on what type of influence on dopaminergic activity exerts NT and NT R ligands depending on their micro- localization, although "Evidence at the molecular level shows the interaction of NTR1 and dopaminergic type 2 receptors forming a heterodimer where NT modulates DA signaling" L552.
L616: The association of NTR1 with dopamine 2 receptors (DA2R) leads to a reduced affinity of DA2R for dopamine in striatopallidal MSNs (GABA medium spiny neurons expressing dopamine receptors) within the NA and cuts alcohol consumption. However, NT also promotes dopamine signaling in the dorsal striatum."
Do the author mean that NTR1 activation promotes D1 mediated dopamine activity in the dorsal striatum (as D1 and D2 receptors are equally represented in this structure)?
L625: "The balance of all the effects at present favors that activation of neurotensinergic systems generally leads to reinforced alcohol responses (narcosis, hypothermia, locomotor activity) with consequent reduction of ethanol drinking by a combined regulation of dopamine and glutamate brain tones [94]."
Do the author mean that NRT1 activity promotes alcohol-driven choice beahvior and lower the ability to acquire habit-driven behavior in preclinical models?
the conclusion of the paragraph L643 "Therefore, the analysis of the implication of NT in the alterations generated by ethanol after acute and chronic exposure needs further and deeper exploration, both in animal models and cell cultures, in vitro analysis of molecular mechanisms, as well as in humans (considering the apparent restrictions) to ascertain its role and open the possibility of using NT analogs to pharmacologically target the receptors to modulate and reverse some symptoms and clinical manifestations of AUD", an acknowledgement of what we do not know is accurate and should be used in final discussion/conclusion of the paper.
NT Receptors Ligands as Possible Pharmacological Treatments of AUD:
Despite the unperfect knowledge of the role or NT and NT R, this paragraph is very interesting, especially the progression from peptidic to non-peptidic stable ligands.
Conclusion and future perspective:
this is a very balanced and cuatious conclusion. the authors should put more emphasis on, as far as I understood, the limitation of NRT1 influence to D2 type DA receptors. Thus, the NT should exert their influence only on the brake and not on the main (D1) way to stimulate plaisure.
the authors aknowledge the risk of modulating NT pathways that have multiple physiological roles, they should also recall the reader that peptidic phamracological agents are hard to deliver in the Human living brain.
Here as a reader, I am not shoked that future treatments of schizophrenia or alcohol abuse are evoked, as the authors write about perspectives.
overall a very interesting,comprehensive and challenging paper that I was very happy to review. I learned a lot and would be happy to have this paper accepted, providing that the authors accept the minor suggestions that I formulated.
Author Response
Thank you for this manuscript's analysis, comments, and constructive criticism. We agree that some changes must be included in the MS to clarify that most results analyzed are sourced from preclinical research. Efforts must be made to understand the impact of neurotensinergic activity in AUD and the possible applications of pharmacological treatments in AUD and other brain diseases.
Please, find our answers to the comments and suggestions and the modifications made in the text of the MS.
-Abstract. We have rephrased as suggested (line 24 and lines 31-32)
- Introduction
The observation that most of the health impacts, particularly deaths, caused by alcohol drinking pertain to severe forms of AUD has been included to support the necessity of specific treatments that may help AUD patients to quit or reduce alcohol drinking and avoid death or further damage. Reference by John U et al. 2022 has been added (lines 62-63).
Paragraph commencing with "Brain peptidergic system… vicious addiction circle has been repositioned (lines 108-114)
Sodium oxybate and topiramate are included and cited (lines 140-141)
- NT and NTR
Some rephasing has been introduced to underly the preclinical evidence supporting the roles of NT (lines 255-256, and lines 265-266)
2.2 Brain distribution of NT
A new paragraph has been added at the end of this section (lines 296-299).
3.1 Brain reward circuit
A new paragraph regarding the relevance of the distinct role of DA 1 and 2 receptors has been added with citations (lines 721-734).
3.3 NT and alcohol
A few changes within the text have been made to clarify the lack of knowledge concerning how NT may influence dopaminergic neurotransmission (lines 834-839 and lines 858-860).
- NT Receptors Ligands as Possible Pharmacological Treatments of AUD
A new table (Table 1) compiles a few examples showing experimental evidence that supports the search for pharmacological strategies to treat AUD and other diseases associated with NT signaling (see page 19, from line 1031).
- Conclusions and Future Perspectives
A short sentence points to the need to unravel the role of NTR1 on dopaminergic neurotransmission in the context of reward and habit-consolidation associated with substance abuse (lines 1048-1051).
The Paragraph: "Therefore, the analysis of the implication of NT in the alterations generated by ethanol after acute and chronic exposure … some symptoms and clinical manifestations of AUD" has been placed at the end of this section, as suggested (lines 1096-1100).
